

# Dental health care providers' concerns, perceived impact, and preparedness during the COVID-19 pandemic in Saudi Arabia

Muhammad Qasim Javed[1], Farooq Ahmad Chaudhary[2], Syed Fareed Mohsin[3], Mustafa Hussein AlAttas[1], Hadeel Yaseen Edrees[4], Syed Rashid Habib[5] and Arham Riaz[6]

[1] Department of Conservative Dental Sciences and Endodontics, College of Dentistry, Qassim University, Buraidah, Qassim, Saudi Arabia
[2] Department of Community Dentistry, School of Dentistry, Shaheed Zulfiqar Ali Bhutto Medical University, Islamabad, Pakistan
[3] Department of Oral Maxillofacial Surgery and Diagnostic Sciences, College of Dentistry, Qassim University, Ar Rass, Qassim, Saudi Arabia
[4] Endodontic Department, Faculty of Dentistry, King Abdulaziz Univeristy, Jeddah, Saudi Arabia
[5] Department of Prosthetic Dental Sciences, College of Dentistry, King Saud University, Riyadh, Saudi Arabia
[6] Community Dentistry, Academy of Continuing Health Education and Research, Islamabad, Pakistan

Corresponding author
Muhammad Qasim Javed,
M.Anayat@qu.edu.sa

## ABSTRACT

**Background.** Dental health care providers (DHCPs)are at high risk of cross-infection during clinical practice therefore, the aim of the study was to evaluate the DHCPs Covid-19 related concerns, its perceived impact, and their preparedness in Saudi Arabia.

**Methods.** This cross-sectional study on DHCPs was carried out at five dental teaching hospitals/colleges in four provinces of Saudi Arabia from October to December 2020. A 35-item valid and reliable questionnaire was used to assess the concerns, perceived impact, and preparedness of DHCPs in the COVID-19 pandemic. Chi-square tests and logistic regression were used to compare parameters between the clinical and non-clinical staff.

**Results.** A total of 320 DHCPs participated in this study with proportion of clinical staff (57.5%) surpassing the non-clinical staff (42.5%). The clinical DHCPs felt greater odds of falling ill with COVID-19 than non-clinical workers (OR, 2.61) and willing to look for another job (OR, 3.50). The higher proportion in both groups was worried that people close to them would be at higher exposure risk (96.3%) however, slightly more clinical DHCPs were concerned for their children than a non-clinical worker (OR, 3.57). The clinical DHCPs have greater odds of worrying that people would avoid them and their family members because of their job (OR, 2.75). A higher proportion in both groups (75.0%, 63.2%) felt that they would feel stress at work. More non-clinical DHCPs (94.1%) had received training for infection control than clinical (94.1% vs 63.0%: OR 0.10). Similarly, more DHCPs in the nonclinical group received adequate personal protective equipment training (88.2%; OR, 0.48). Most participants practiced self-preparation such as buying masks and disinfection (94.4%, 96.9%).

**Conclusion.** The majority of DHCPs felt concerned about their risk of exposure and falling ill from infection and infecting friends/family. These concerns could potentially affect the working of DHCPs during this pandemic. Measures to improve protection

for DHCPs, minimize psychological implications, and potential social stigmatization should be identified at the planning phase before any pandemic.

# INTRODUCTION

By the end of 2019, the COVID-19 outbreak in Wuhan, China made global headlines and continued to spread globally at a rapid speed (*Zhu, Wei & Niu, 2020*). The World Health Organization (WHO) had declared the COVID-19 outbreak, a pandemic on the 11th of March 2020 (*Cucinotta & Vanelli, 2020*). Currently, the virus has affected 221 countries and territories with approximately, 102 million individuals have been infected with the virus worldwide, with a mortality rate of 2.1% (*World Health Organization, 2020*). The Saudi Ministry of Health reported the first COVID-19 case on 2nd March 2020, when an individual returning from Iran was tested positive (*Ministry of Health, 2020*). As of 1st February 2021, the Kingdom of Saudi Arabia has reported about 368,074 confirmed cases. The total recovered patients are 359,573, with 6375 deaths in the kingdom. Overall, the positive cases to PCR test, (12,295,687 tests) ratio are around 3% (*Ministry of Health, 2020*).

It is highly communicable pathogen that can be easily transmitted via virus-laden respiratory droplets of the infected individual (*Khan, Nawabi & Javed, 2020*). In this context, National Health Services England and American Dental Association issued guidelines for dental health care providers (DHCPs) and suggested that all elective dental treatments should be postponed and only patients with dental emergencies should be accommodated (*National Health Services England, 2020*; *The American Dental Association, 2020*). With the total sum of patients who have contracted COVID-19 disease approaching 102 million globally, it is almost certain that some of the infected individuals will need emergency dental care (*World Health Organization, 2020*). Considering that the evidence supporting the use and effectiveness of antibiotics for acute dental conditions (pulpal and periapical pain) is weak (*Lockhart et al., 2019*), it seems that DHCPs will have to carry out the acute dental care for the known or suspected COVID-19 patients. The guidelines for the dental management of COVID-19 patients are continuously evolving and the outbreak has placed DHCPs at a very high risk of acquiring nosocomial infections. The bio-aerosols produced during dental procedures have the potential to float in the air for a considerable period (*Occupational Safety and Health Administration, 2020*). A recently published review has discussed in detail the necessary requirements of personal protective equipment (PPE), administrative, and environmental control for the provision of acute dental care in the current circumstances (*Ather et al., 2020*). It is recommended that suspected or confirmed COVID-19 patients should not be treated in the neutral pressure rooms of routine dental practices. Instead, the aforementioned patient group should only be treated in airborne infection isolation rooms or negative pressure treatment rooms (*Centers for Disease Control and Prevention, 2020*).

With the increased risk of exposure to DHCPs due to the close contact with the COVID-19 infected patients' oropharyngeal region and distinctive features of dental treatment procedures that involve aerosol generation, routine infection control measures in dental practice are not adequate to prevent the COVID-19 spread (*Kohn et al., 2003*). Consequently, Dental healthcare facilities have made the use of PPE mandatory while treating patients. The enhanced exposure risk among healthcare workers (HCWs), results in stress and fear of contracting the infection (*Dubey et al., 2020*; *Boyraz & Legros, 2020*). A study in Poland showed that 71.2% of the dentists voluntarily suspended their practice due to fear, anxiety and unpreparedness of oral health care sector both at public and private settings for COVID-19 outbreak (*Tysiąc-Miśta & Dziedzic, 2020*). Likewise, in Australia majority (90–93%) oral health care staff and students in Dental hospital perceived their health to be at risk and this increased their stress and impacted their clinical performance (*Loch et al., 2021*). According to Amnesty international COVID-19 disease has resulted in the death of around 7000 healthcare professionals, globally with lack of PPE as one of the contributory factors (*Amnesty International, 2020*).

In Saudi Arabia, several HCWs have lost their lives as a result of COVID-19 disease complications and many more suffering from psychological morbidities (*Arab News, 2020*). Previous reports on COVID-19 infection related concerns and its impact involved the medical health care workers only, little is understood about the oral health care workers despite facing similar challenges and occupational hazards. Only one study in Pakistan reported the dental health care providers COVID-19 related concerns, its impact and preparedness in dental hospitals (*Wong et al., 2008*; *Matsuishi et al., 2012*). More studies are needed to be carried out to understand the difference in concerns, perceived impact and preparedness of DHCPs working in different socioeconomics and geographical regions. Therefore, the objective of the study was to evaluate the perceived impact, preparedness, and concerns of dental healthcare providers during the COVID-19 disease pandemic in the dental hospitals of Saudi Arabia

## MATERIALS AND METHODS

The current cross-sectional study was conducted at five dental teaching hospitals/colleges in four provinces (Qassim, Riyadh, Jeddah, Eastern Province) of the Kingdom of Saudi Arabia from October to December 2020. The ethical approval for this research was taken from the Dental Ethics Committee of Qassim University (Reference Code: ST/6080/2020). A convenient sampling technique was used to recruit the respondents. All the participants voluntarily participated in the study; the identified participants were contacted by an appointed staff at the respective institutions/hospital. The purpose of the study was explained by sharing the information sheet with DHCPs at the dental hospitals. A written informed consent was obtained, and participants were requested to self-administered the study questionnaire. The first part of the questionnaire collected information regarding socio-demographic characteristics of the participants which included age, gender, marital status, place of work, and place of living. A valid and reliable questionnaire consisting of 35-items in four sections was adopted from *Chaudhary et al. (2020)* to assess the concerns,

perceived impact, and preparedness of DHCPs in the COVID-19 pandemic. The concerns related to work and social life of DHCPs were assessed in the first two sections of the adopted questionnaire and perceived impact and preparedness of DHCPs was assessed in the last two sections. The participants recorded their responses on a 6-point Likert scale ranging from 1 (strongly agree, agree, and probably agree) to 6 (strongly disagree, disagree, and probably disagree). The responses were dichotomized, former 3 responses were recoded as agree and the latter 3 as disagree for the purpose of analysis. The DHCPs were classified into two subgroups: non-clinical (basic sciences faculty, managerial/clerical staff, dental lab technician, attendant/cleaners) and clinical (faculty and consultants of clinical subjects, general dentists, dental hygienists, and assistants) depending upon their direct interaction with the patients.

### Statistical analysis

SPSS version 23 (SPSS Institute, Chicago, IL, USA) was used for the statistical analysis. Descriptive statistics were calculated for all variables. The chi-square tests were used to evaluate the association between socio-demographics variables with clinical and non-clinical staff. Binary logistic regression adjusted for socio-demographics variables were used to compare the parameters between non-clinical and clinical staff. The level of significance was set at < 0.05.

## RESULTS

The questionnaires were distributed among 350 subjects however only 320 (91.4%) participants were able to complete all the questions. A slight exceeding number of Clinical staff ($n = 184$, 57.5%) participated in the study as compared to non-clinical staff ($n = 136$, 42.5%). The majority of participants in both the groups were below the age of 40 years (54%, 94.1%), A significantly higher proportion of non-clinical male staff was found than clinical staff ($p < 0.001$). Similarly, the majority of participants in both the groups were working in the government sector (87.3%), and significantly higher proportion of non-clinical staff living with friends and family compared to clinical staff ($P < 0.001$). The demographic characteristics of dental healthcare providers are shown in Table 1.

A higher proportion of clinical DHCPs were afraid of falling ill with COVID-19 than non-clinical workers (90.2% vs 77.9%; OR, 2.61; 95% CI [1.38–4.91]). However, 69.6% of DHCPs in this study think that they should not be looking after COVID-19 patients and only 57.5% of DHCPs felt that this risk of exposure is not acceptable. However, a majority (80.4%) of them accepted the risk of contracting COVID-19 as part of the job. Very small proportions from both groups were looking for another job because of this risk (7.4%), but more clinical DHCPs were willing to look for another job than non-clinical DHCPs (21.7% vs 11.0%; OR, 3.50; 95% CI [1.68–7.28]), and a minority of them think that it's acceptable if colleagues resign because of the fear (42.5%) (Table 2).

The higher proportion in both groups were worried that people close to them would be at higher exposure risk (96.3%), and more concerned were related to spouse/partner, parents, close friends, and work colleagues in both groups (90.0%, 96.9%, 97.5%, and

| Table 1 Demographic characteristics of dental healthcare professionals. | | | |
|---|---|---|---|
| | Clinical staff N(%) N = 184(57.5) | Non-clinical staff N(%) N = 136(42.5) | Chi-square (P-value) |
| Age | | | |
| 20–29 | 34 (18.5) | 128 (94.1) | |
| 30–39 | 64 (34.8) | 0 | |
| 40–49 | 62 (33.7) | 6 (4.4) | 0.001 |
| 50–60 | 16 (8.7) | 0 | |
| 60–70 | 8 (4.3) | 2 (1.5) | |
| Gender | | | |
| Male | 110 (60.0) | 110 (81.0) | 0.001 |
| Female | 74 (40.2) | 26 (19.1) | |
| Civil status | | | |
| Single | 44 (23.9) | 118 (86.8) | |
| Married | 136 (73.9) | 18 (13.2) | 0.001 |
| Divorced | 4 (2.2) | 0 | |
| Place of work | | | |
| Government/public-Sec. | 154 (83.7) | 136 (100.0) | 0.001 |
| Private sector | 30 (16.3) | 0 | |
| Staying with | | | |
| Family/friend | 136 (73.9) | 128 (94.1) | 0.001 |
| Alone | 48 (26.1) | 8 (5.9) | |
| Location | | | |
| Qasim | 92 (50.0) | 82 (60.3) | |
| Riyadh | 12 (6.5) | 22 (16.2) | 0.001 |
| Jeddah | 42 (22.8) | 16 (11.8) | |
| Eastern Province | 38 (20.7) | 16 (11.8) | |

95.0%). However, slightly more clinical workers were concerned for their children than a non-clinical worker (97.8% vs 92.6%; OR, 3.57; 95% CI [1.09–11.6]) (Table 3).

More clinical workers were afraid that people would avoid them and their family members because of their job than non-clinical workers (65.2 vs 44.1; OR, 2.37; 95% CI [1.50–3.74] and 47.8 vs 25.0; OR, 2.75; 95% CI [1.69–4.46]). More than half of the participants (55.6%) in both groups felt that there is an inadequate staff at their workplace to handle increase demand and similarly the same percentage of participants in both the groups (55.6%) felt that there would be more conflict among colleagues at their workplace. A higher proportion of clinical workers (75.0%) than non-clinical workers (63.2%) think that they would feel stress at work (OR, 1.74; 95% CI [1.07–2.82]) (Table 4).

Participants in both the groups (87.5%) know that there is an infection control committee & staff in the hospital, but more non-clinical workers (94.1%) had received training for infection control at the hospital than clinical workers (94.1% vs 63.0%; OR 0.10; 95% CI [0.04–0.23]). Similarly, more nonclinical workers knew about preparedness plan for Covid-19 outbreak than clinical workers (80.9% vs 64.1%, OR 0.42: 95% CI [0.25–0.71]).

**Table 2  COVID-19 pandemic and work-related concerns of dental healthcare professionals.**

| Concerns (Agree) | Clinical N (%) | Non-clinical N (%) | Total N (%) | Unadjusted OR | P-value | Adjusted OR[a] | P-value |
|---|---|---|---|---|---|---|---|
| Work-related concerns | | | | | | | |
| My job would put me at great exposure risk | 166(90.2) | 130(95.6) | 296(92.5) | 0.12 (0.02–0.57) | 0.007 | 0.42 (0.16–1.10) | 0.07 |
| I am afraid of falling ill with Covid-19 | 166(90.2) | 106(77.9) | 272(85.0) | 2.16 (0.80–5.80) | 0.12 | 2.61 (1.38–4.91) | 0.003 |
| I should not be looking after Covid-19 patients | 128(69.6) | 68(50.0) | 196(61.3) | 3.66 (1.68–7.95) | 0.001 | 2.28 (1.44–3.62) | 0.001 |
| The risk I am exposed to is not acceptable | 116(63.0) | 68(50.0) | 184(57.5) | 1.49 (0.75–2.98) | 0.251 | 1.70 (1.08–2.67) | 0.020 |
| I accept that risk of contracting Covid-19 is part of job | 148(80.4) | 128(94.1) | 276(86.3) | 0.43 (0.15–1.24) | 0.119 | 0.25 (0.11–0.57) | 0.001 |
| Might look for another job because of risk | 40(21.7) | 10(11.0) | 50(7.4) | 3.77 (1.40–10.1) | 0.008 | 3.50 (1.68–7.28) | 0.001 |
| Acceptable if colleagues resign because of their fear | 74(40.2) | 62(45.6) | 136(42.5) | 0.68 (0.34–1.36) | 0.28 | 0.80 (0.51–1.25) | 0.33 |
| Health care employers would look after my needs if I fall ill with Covid-19 | 140(76.1) | 106(77.9) | 246(76.9) | 1.27 (0.59–2.73) | 0.532 | 0.90 (0.53–1.52) | 0.69 |

Notes.
[a] Adjusted for age, gender, civil status, place of work, "Staying with".

**Table 3  COVID-19 pandemic and non-work related concerns of dental healthcare professionals.**

| Concerns (Agree) | Clinical N (%) | Non-clinical N (%) | Total N (%) | Unadjusted OR | P-value | Adjusted OR[a] | P-value |
|---|---|---|---|---|---|---|---|
| Non-work concerns | | | | | | | |
| People close to me would be at high risk of getting Covid-19 because of my job | 174 (94.6) | 134 (98.5) | 308 (96.3) | 0.55 (0.07–4.10) | 0.56 | 0.26 (0.56–1.20) | 0.085 |
| *I would be concerned for my:* Spouse/partner | 174 (94.6) | 114 (83.8) | 288 (90.0) | 1.84 (0.58–5.83) | 0.29 | 3.35 (1.53–7.35) | 0.002 |
| Parents | 176 (95.7) | 134 (98.5) | 310 (96.9) | 0.76 (0.12–4.76) | 0.77 | 0.32 (0.06–1.57) | 0.16 |
| Children | 180 (97.8) | 126 (92.6) | 306 (95.6) | 3.39 (0.68–16.9) | 0.13 | 3.57 (1.09–11.6) | 0.035 |
| Close friends | 176 (95.7) | 136(100.0) | 312 (97.5) | – | - | – | - |
| Work colleagues | 176 (95.7) | 128 (94.1) | 304 (95.0) | 0.52 (0.12–2.26) | 0.38 | 1.37 (0.50–3.76) | 0.53 |
| People close to me would be worried for my health | 178 (96.7) | 132 (97.1) | 310 (96.9) | 0.56 (0.10–3.10) | 0.508 | 0.89 (0.24–3.25) | 0.87 |
| People close to me would be worried as they may get infected by me | 178 (96.7) | 132 (97.1) | 310 (96.9) | 0.56 (0.10–3.10) | 0.50 | 0.89 (0.24–3.25) | 0.87 |

Notes.
[a] Adjusted for age, gender, civil status, place of work, "Staying with".

**Table 4  Dental healthcare professionals' perceived impact on work and personal life.**

| Perceived impact (agree) | Clinical N = % | Non-clinical N = % | Total N = % | Unadjusted OR | P-value | Adjusted OR[a] | P-value |
|---|---|---|---|---|---|---|---|
| I would be afraid of telling my family/friends about the risk I am exposed | 90(48.9) | 64(47.1) | 154(48.1) | 1.75(0.88–3.49) | 0.10 | 1.07(0.69–1.67) | 0.74 |
| People would avoid me because of my job | 120(65.2) | 60(44.1) | 180(56.3) | 4.50(2.54–7.97) | 0.001 | 2.37(1.50–3.74) | 0.001 |
| People would avoid my family members because of my job | 88(47.8) | 34(25.0) | 122(38.1) | 2.44(1.21–4.89) | 0.012 | 2.75(1.69–4.46) | 0.001 |
| I would avoid telling other people about the nature of my job | 48(26.1) | 22(16.2) | 70(21.9) | 7.79(3.18–19.0) | 0.001 | 1.82(1.04–3.21) | 0.035 |
| There would be inadequate staff at my workplace to handle the increased demand | 120(65.2) | 58(42.6) | 178(55.6) | 2.66(1.29–5.48) | 0.008 | 2.52(1.59–3.97) | 0.001 |
| There would be more conflict amongst colleagues at work | 108(58.7) | 70(51.5) | 178(55.6) | 1.37(0.70–2.67) | 0.34 | 1.34(0.85–2.09) | 0.19 |
| I would feel more stressed at work | 138(75.0) | 86(63.2) | 224(70.0) | 4.54(2.0–10.2) | 0.001 | 1.74(1.07–2.82) | 0.024 |
| I would have an increase in workload | 134(72.8) | 88(64.7) | 222(69.4) | 2.38(1.12–5.06) | 0.024 | 1.46(0.90–2.35) | 0.12 |
| I would have to do work not normally done by me | 108(58.7) | 68(50.0) | 176(55.0) | 1.47(0.75–2.90) | 0.025 | 1.42(0.91–2.20) | 0.12 |

Notes.
[a] Adjusted for age, gender, civil status, place of work, "Staying with".

Participants in both the groups bought disinfection equipment (94.4%) and masks (96.9%). However, more DHCPs in the nonclinical group received adequate personal protective equipment training than clinical DHCPs (88.2% vs 78.3%; OR, 0.48; 95% CI [0.25–0.90])(Table 5).

## DISCUSSION

This study investigates the perceived impact and preparedness of dental health care providers during the COVID-19 pandemic in Saudi Arabia. Greater percentage of participants was from Al Qasim region (54.3%) as two participating Dental teaching institutes/hospitals were from Qassim region. Moreover, there are some other factors that might have resulted in the high response rate from Qassim region, such as comprehensive COVID 19 related online awareness/training program for the staff members and students organized by the infection control unit at college of Dentistry and multiple reminders that were sent to the potential participants.

This study reports that a higher percentage of clinical DHCPs were afraid of falling ill with COVID-19 than non-clinical workers. Our finding is in line with other studies on

**Table 5  Preparedness of dental healthcare professionals for COVID-19 pandemic.**

| Statement (agree) | Clinical N (%) | Non-clinical N (%) | Total N (%) | Unadjusted OR | P-value | Adjusted OR[a] | P-value |
|---|---|---|---|---|---|---|---|
| There is an infection control committee & staff in my hospital | 154(83.7) | 126(92.6) | 280(87.5) | 0.22(0.08–0.64) | 0.005 | 0.40(0.19–0.86) | 0.019 |
| I have received training for infection control at my hospital | 116(63.0) | 128(94.1) | 244(76.3) | 0.10(0.037–0.27) | 0.001 | 0.10(.049-0.23) | 0.001 |
| My clinic has a preparedness plan for a Covid-19 outbreak | 118(64.1) | 110(80.9) | 228(71.3) | 0.50(0.23–1.06) | 0.074 | 0.42(0.25–0.71) | 0.001 |
| My hospital has informed me of their Covid-19 outbreak preparedness plan | 116(63.0) | 90(66.2) | 206(64.4) | 1.13(0.56–2.30) | 0.718 | 0.87(0.54–1.38) | 0.56 |
| I am personally prepared for a Covid-19 outbreak | 136(73.9) | 122(89.7) | 258(80.6) | 0.68(0.29–1.62) | 0.39 | 0.32(0.17–0.61) | 0.001 |
| In the past 6 months | | | | | | | |
| I have attended infection control training sessions | 88(47.8) | 90 (66.2) | 178(55.6) | 0.54(0.27–1.08) | 0.082 | 0.46(0.29–0.74) | 0.001 |
| Bought disinfection | 174(94.6) | 128(94.1) | 302(94.4) | 0.72(0.17–2.97) | 0.65 | 1.08(0.41–2.83) | 0.86 |
| Bought masks | 180(97.8) | 128(95.5) | 308(96.9) | 0.81(0.10–6.18) | 0.84 | 2.10(0.58–7.62) | 0.25 |
| Received adequate personal protective equipment training | 116(63.0) | 112(82.4) | 228(71.3) | 0.51(0.23–1.11) | 0.09 | 0.36(0.21–0.62) | 0.001 |
| Have someone to turn to if unsure of use of personal protective equipment | 144(78.3) | 120(88.2) | 264(82.5) | 1.65(0.67–4.07) | 0.27 | 0.48(0.25–0.90) | 0.022 |

Notes.

[a] Adjusted for age, gender, civil status, place of work, "Staying with".

HCWs in Singapore during the Avian Influenza outbreak and oral healthcare workers in Pakistan (*Wong et al., 2008*; *Chaudhary et al., 2020*).

Since it has been confirmed that the main route for coronavirus transmission is through droplets, the probability of oral healthcare workers being infected is increasing (*Ge et al., 2020*). In this study majority of clinical DHCPs (70%) think that they should not be looking after COVID-19 patients, however, a higher proportion of DHCPs (86.3%) felt that contracting COVID-19 infection is a part of the job. These findings are in agreement with previous studies on health workers where the majority agreed that contracting an infection during the job is part of their profession and they are at high risk of contracting the virus. The plausible explanations may be related to a higher awareness of DHCPs provided by social media networks about high-risk factors, their outcome, other factors like shortages of personal protective equipment (PPE) in hospitals, extended workloads, insufficient testing, and other emerging issues (*Chaudhary et al., 2020*).

A small number of DHCPs (7.4%) reported that they would look for another job. The response is consistent with other studies conducted in Singapore, the USA, and Pakistan, where most healthcare workers agreed to continue work during SARS, Avian influenza, and COVID-19 outbreak respectively (*Chaudhary et al., 2020*; *Wong et al., 2008*; *Martin,*

*2011*). In contrast, other studies reported a higher proportion of healthcare workers from Taiwan, Hong Kong, and the United Kingdom (43%–77%) refused to work during an outbreak and preferred to look for another job or quit their job (*Chaudhary et al., 2020*; *Wong et al., 2010*; *Shiao et al., 2007*; *Barr et al., 2008*).

A significant proportion of participants in both groups responded that people close to them would be at higher exposure risk. In contrast, clinical workers were more concerned about exposure risk to their children than non-clinical workers. These findings were observed during the SARS outbreak as well, where media reported discrimination against HCWs and their family members (*Khee et al., 2004*; *Tai, 2006*). In the present COVID-19 pandemic, many countries effectively highlighted the dedication and sacrifices offered by HCWs (*Tang, 2020*). This strategy has changed public opinion resulting in supporting their effort of serving the people and the country. Law enforcement agencies in Pakistan have also gave the Guard of Honour and one month honorarium as a token of appreciation to HCWs for their hard work, dedication and contribution as a front liner in the battle against COVID-19 (*News, 2020*). However, the results of this study indicate that the perception and fear of prejudice against health workers are still sustained, and more efforts should be done to change people's perceptions.

More than half of the respondents reported that there was inadequate clinical staff to handle increasing demand. Moreover, a higher proportion of clinical workers were under stress when compared to non-clinical workers. These findings are consistent with other studies where hospitals were overwhelmed with the patients and insufficient health staff was working extra hours to handle this load. This workload along with fear of exposure to COVID-19 to them and their family members made health workers more stressed and worried as shown in this study (*Chaudhary et al., 2020*).

Both clinical and non-clinical DHCPs involved in infection control activities and seemed to be more aware of their preparedness plan however, a greater proportion of non-clinical DHCPs (66.2%) attended infection control training sessions recently than clinical DHCP (47.8%). This result is in contrast with the study in Pakistan where more clinical oral health workers attended infection control training sessions than non-clinical oral health workers. One of the plausible reason for this disparity may be that the personal preparedness of clinical oral health workers in Saudi Arabia is much better (73.9%) compared to clinical oral health workers in Pakistan (30.9%) and secondly, the majority of DHCPs (87.5%) in this study knew about the working of infection control committee in the hospital than in Pakistan (51.3%) (*Wong et al., 2008*; *Chaudhary et al., 2020*). The DHCPs are at high risk for this disease and therefore need sufficient training not just as precautionary but also to show confidence and high morality while at work. Many countries have established criteria for provision of oral care services during this pandemic, including training of control infections and establishing safety protocols that should be part of infection control in dental hospitals (*Ahmed et al., 2020*).

The WHO has highlighted the importance of preparedness plans to mitigate the adverse impact of this pandemic and like many other countries Saudi Arabia have also prepare and implement their national pandemic Covid-19 preparedness plan based on WHO guidelines (*MOH, 2020*; *WHO, 2021*). These guidelines ensuring adequate support for frontline health

and oral health care workers, including access to prevention, treatment, and provision of medications or vaccines. The breakthrough in this pandemic is the invention of the vaccine and many countries, including Saudi Arabia, have started administering the COVID-19 vaccine as the priority to frontline health and oral health care workers and people with chronic diseases to combat this deadly disease and keep high hope to control this infection. Moreover, in case of financial constraints because of the exponential spread of COVID-19 disease the infection control for the DHCPs' can be achieved by implementing the frugal solutions (*Javed & Bhatti, 2020*).

The cross-sectional study design and use of self-administered questionnaires have their limitations such as recall, framing, and rating bias. This research's main strength was that it was the first study that explored the concerns, impacts, and preparedness of DHCPs in Saudi Arabia during the COVID-19 pandemic. This study will help to contribute to the development of further studies on DHCPs related to COVID-19.

## CONCLUSION

This study revealed that DHCP are extremely vulnerable to the risk of the COVID-19 infection resulting in infecting family and friends due to their profession. The majority of DHCPs were well prepared for this pandemic both personally and at the workplace. The concerns observed in this study could ultimately affect the overall efficiency of oral health care workers during a pandemic. Measures to improve protection for DHCPs, minimize psychological implications, and potential social stigmatization should be identified at the planning phase before any pandemic.

### Funding
The Deanship of Scientific Research, Qassim University funded the publication of this project. The funders had no role in study design, data collection and analysis, decision to publish, or preparation of the manuscript.

### Grant Disclosures
The following grant information was disclosed by the authors:
Deanship of Scientific Research, Qassim University.

### Competing Interests
The authors declare there are no competing interests.

### Author Contributions
- Muhammad Qasim Javed conceived and designed the experiments, performed the experiments, analyzed the data, prepared figures and/or tables, authored or reviewed drafts of the paper, and approved the final draft.
- Farooq Ahmad Chaudhary conceived and designed the experiments, analyzed the data, prepared figures and/or tables, authored or reviewed drafts of the paper, and approved the final draft.

- Syed Fareed Mohsin, Mustafa Hussein AlAttas and Hadeel Yaseen Edrees performed the experiments, authored or reviewed drafts of the paper, and approved the final draft.
- Syed Rashid Habib analyzed the data, authored or reviewed drafts of the paper, and approved the final draft.
- Arham Riaz conceived and designed the experiments, authored or reviewed drafts of the paper, and approved the final draft.

## Human Ethics

The following information was supplied relating to ethical approvals (i.e., approving body and any reference numbers):

The Dental Ethics Committee, College of Dentistry, Qassim University, Saudi Arabia approved this research (Ethical Approval Reference number: ST/6080/2020).

## Data Availability

The raw data are available in the Supplementary Files.

## Supplemental Information

Supplemental information for this article can be found online at http://dx.doi.org/10.7717/peerj.11584#supplemental-information.

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
