# Peer review of "Dental health care providers' concerns, perceived impact, and preparedness during the COVID-19 pandemic in Saudi Arabia"

_PeerJ, doi:10.7717/peerj.11584_

## Round 0.1 · original submission · Major Revisions

As you can see, the recommendations of the two reviewers were dramatically different. I believe that if you make it clear that you used a convenience sample in your methodology that your methods are appropriate. When you revise your manuscript, be sure to address ALL of the concerns of the two reviewers. Specifically, it is important that you address the impression that you have no clear methodology in your study.

·

Basic reporting

Dental health care providers' concerns, perceived impact, and preparedness during the COVID-19 pandemic in Saudi Arabia

Abstract
Background – Please include the study objective.
Method – Please include the variables measured and the specific analysis conducted – what parameters were compared.
Findings – How many clinical and non-clinical workers were recruited? May be better to first report the descriptive variables before the inferentials
Conclusion – does not discuss perceived impact, and preparedness in a COVID-19 pandemic. I guess because it was not measured as this is not in the results.

Introduction
The opening sentence does not introduce the manuscript. This manuscript is about COVID-19 and not coronavirus in humans. Please revise the opening sentence
There is no clear justification for the study. Why is there a need to conduct the study in Saudi Arabia?
There is no theoretical framework informing the study or a conceptual framework that helps one to identify the variables to be studied. Why clinicians and non-clinicians for this study? The introduction is not focused.

Experimental design

Methodology
Please use the STROBE guideline to develop a study methodology. Right now the study methodology is missing. In the absence of this, it is not possible to evaluate the results and discussion

Validity of the findings

In the absence of a methodology, the results cannot be assessed

·

Basic reporting

Javed et al., Dental Health Care Providers concerns, perceived impact and preparedness during the COVID =019 pandemic in Saudi Arabia.

Manuscript presents results of a survey of clinical and non-clinical facing staff at five hospital University clinics in Saudi Arabia made in fall 2020. The general purpose was to assess qualitative stress and attitudes of the two groups. In general this is an informative study if not limited to just Saudi Arabia but the results can be informative in creating communication policies and programs.

Experimental design

Some specific comments are:
• With a convenience response sample of 320 it would be good to know the total response rate (how many were approach to complete the survey)?
• Second, it would be reasonable to either plot the distribution of Likert responses to determine homo/heterogeneity of sample responses since the dichotomization approach used and subsequent nonparametric chi-square suggests heterogeneity in the frequency and distribution of scores.
• While the response rate may vary site to site between the five centers, it would be reasonable to outline if one site had more responses and if that regional site had more news coverage, local outbreak or other local social demographic explanatory variable for the responses recorded.
• A smaller issue is the use of new media as cited sources (ref 14 and 15) can be problematic as the veracity of these citations is unclear.
• The references may need to be up dated with the more recent studies indicating low transmission in the dental office setting.

Ren Y, Feng C, Rasubala L, Malmstrom H, Eliav E. Risk for dental healthcare professionals during the COVID-19 global pandemic: An evidence-based assessment. J Dent. 2020 Oct;101:103434. doi: 10.1016/j.jdent.2020.103434. Epub 2020 Jul 18. PMID: 32693111; PMCID: PMC7368403.

Tysiąc-Miśta M, Dziedzic A. The Attitudes and Professional Approaches of Dental Practitioners during the COVID-19 Outbreak in Poland: A Cross-Sectional Survey. Int J Environ Res Public Health. 2020 Jun 30;17(13):4703. doi: 10.3390/ijerph17134703. PMID: 32629915; PMCID: PMC7370196.

Jum'ah AA, Elsalem L, Loch C, Schwass D, Brunton PA. Perception of health and educational risks amongst dental students and educators in the era of COVID-19. Eur J Dent Educ. 2020 Nov 14:10.1111/eje.12626. doi: 10.1111/eje.12626. Epub ahead of print. PMID: 33188555; PMCID: PMC7753269.

Loch C, Kuan IBJ, Elsalem L, Schwass D, Brunton PA, Jum'ah A. COVID-19 and dental clinical practice: Students and clinical staff perceptions of health risks and educational impact. J Dent Educ. 2021 Jan;85(1):44-52. doi: 10.1002/jdd.12402. Epub 2020 Sep 10. PMID: 32914437.

Validity of the findings

In general this is an informative study if not limited to just Saudi Arabia but the results can be informative in creating communication policies and programs.

---

## Round 0.2 · accepted · Accept

Thank you for addressing concerns of the reviewers.

·

Basic reporting

Satisfactory

Experimental design

Satisfactory

Validity of the findings

SAtisfactory